# Actin Isovariant ACT7 Modulates Root Thermomorphogenesis by Altering Intracellular Auxin Homeostasis

**DOI:** 10.3390/ijms22147749

**Published:** 2021-07-20

**Authors:** Sumaya Parveen, Abidur Rahman

**Affiliations:** 1United Graduate School of Agricultural Sciences, Iwate University, Morioka 020-8550, Japan; sumaya.parveen71135@gmail.com; 2Department of Plant Bio Sciences, Faculty of Agriculture, Iwate University, Morioka 020-8550, Japan; 3Agri-Innovation Center, Faculty of Agriculture, Iwate University, Morioka 020-8550, Japan

**Keywords:** high temperature, root elongation, *ACT7*, auxin, actin, protein trafficking

## Abstract

High temperature stress is one of the most threatening abiotic stresses for plants limiting the crop productivity world-wide. Altered developmental responses of plants to moderate-high temperature has been shown to be linked to the intracellular auxin homeostasis regulated by both auxin biosynthesis and transport. Trafficking of the auxin carrier proteins plays a major role in maintaining the cellular auxin homeostasis. The intracellular trafficking largely relies on the cytoskeletal component, actin, which provides track for vesicle movement. Different classes of actin and the isovariants function in regulating various stages of plant development. Although high temperature alters the intracellular trafficking, the role of actin in this process remains obscure. Using isovariant specific vegetative class actin mutants, here we demonstrate that ACTIN 7 (*ACT7*) isovariant plays an important role in regulating the moderate-high temperature response in Arabidopsis root. Loss of *ACT7*, but not *ACT8* resulted in increased inhibition of root elongation under prolonged moderate-high temperature. Consistently, kinematic analysis revealed a drastic reduction in cell production rate and cell elongation in *act7-4* mutant under high temperature. Quantification of actin dynamicity reveals that prolonged moderate-high temperature modulates bundling along with orientation and parallelness of filamentous actin in *act7-4* mutant. The hypersensitive response of *act7-4* mutant was found to be linked to the altered intracellular auxin distribution, resulted from the reduced abundance of PIN-FORMED *PIN1* and *PIN2* efflux carriers. Collectively, these results suggest that vegetative class actin isovariant, *ACT7* modulates the long-term moderate-high temperature response in Arabidopsis root.

## 1. Introduction

High temperature stress is one of the major environmental challenges that limits plant growth and productivity world-wide. Global average temperature has already risen by roughly 0.13 °C per decade since 1950 resulting in increasing crop yield loss [1]. From 1964 to 2007, 7.6% of cereal yield declined due to Extreme Weather Disasters (EWDs) of extreme heat [2]. Heat stress causes the combined production loss of wheat, maize, and barley of around 5 billion USD annually [3]. According to the fifth assessment report of IPCC, global average temperature will be at least 1.5° C higher by the end of the 21st century [4]. To combat the drastic effect of high temperature on crop production, understanding the molecular and cellular mechanisms of high temperature stress responses in plants is essential. 

Depending on the intensity of high temperature treatment and developmental stages, plants show differential thermoresponsive growth. High temperature stress during seed germination reduces germination rate by increasing abscisic acid (ABA) biosynthesis and inhibiting gibberellin (GA) biosynthetic genes [5,6]. Phytochrome B and D (PHY B and D) have been shown to regulate high temperature-induced secondary seed dormancy [7]. High temperature also affects reproductive organs destructively, for example, flower and flower bud abortion [8], reduced pollen production and fruit dehiscence. High temperature stress (33 °C) suppresses auxin level in the anther by reducing the expression of auxin biosynthetic genes *YUCCA2* and *YUCCA6* that results in inhibition of pollen production. This incident could be reversed by exogeneous application of auxin [9]. Exposure to the moderate-high temperature stress (28–30 °C) at the seedling stage results in some adaptive thermomorphogenesis responses such as floral transition, leaf petiole, and hypocotyl elongation, leaf hyponasty, and root elongation [10]. High temperature stress-induced promotion of flowering under short photoperiod is linked to the upregulation of the flowering promoting gene FLOWERING LOCUS T(FT) regulated by Phytochrome Interacting Factor (PIF4) [11].

GA, auxin, and brassinosteroid contribute to the hypocotyl and leaf petiole elongation under high temperature stress [12]. Auxin biosynthesis and signaling mutants, *pif4* and *tir1–1afb2-3afb3-4* show defective high temperature mediated hypocotyl elongation which could be rescued by exogenous brassinosteroid application [13]. High temperature-induced increase in auxin biosynthesis has been attributed to the thermoresponsive transcription factor Phytochrome interacting factor PIF4, which binds to the promoter of auxin biosynthetic genes *YUC8*, *TAA1*, and *CYP79B2* and enhances their expression. The resulting alteration in the intracellular auxin level promotes the leaf and hypocotyl hyponasty [14,15]. Although the thermoresponsive mechanisms in hypocotyl and flower development have been extensively studied and found to be centered on PIF mediated auxin biosynthesis and homeostasis, the mechanism in root is still elusive.

It has been shown that the short-term moderate-high temperature stress increases the intracellular auxin level and stimulates the root elongation [16]. Both the auxin transport and signaling have been implicated in regulating the root growth response under short-term moderately high temperature. Auxin receptor mutant *tir1-1afb2-3* showed reduced root elongation stimulation under high-temperature stress [17]. Similarly, compared to wild-type, auxin transport mutants Ethylene Insensitive in Root mutant (*eir1-1/pin2*) and Auxin Resistant mutant *aux1-7* showed a reduced response to high temperature stress for root elongation [16]. The intracellular auxin homeostasis under moderate-high temperature stress was shown to be linked to the sorting nexin 1 (SNX1)-mediated recycling of PIN formed 2 (*PIN2*) proteins from vacuole to plasma membrane [16]. High temperature-mediated alteration of protein trafficking was also found to be needed for the export of FT protein from companion cells to sieve elements [18]. Misfolding of N470D LQT2 mutant protein at 37 °C leads to defective trafficking in HEK293 cells [19]. Collectively, these results suggest that protein trafficking may play an important role in short-term high temperature-mediated developmental regulation. However, the long-term effect of moderate high temperature on root development and its mechanism remains elusive.

For the intracellular protein trafficking, the cell cytoskeleton component actin plays an important role as it provides track for various types of trafficking that are essential for cellular, developmental, and reproductive processes such as establishing and maintaining cell shape and polarity, tip growth, cytoplasmic streaming and organelle movement, cell division, and cell elongation [20,21]. Intracellular trafficking of auxin efflux carriers such as *PIN1* and *PIN2* that regulate the polar transport of auxin has also been shown to be actin dependent [22,23,24]. Because high temperature alters the intracellular trafficking of *PIN2* [16], and since *PIN2* trafficking is actin dependent, we hypothesized that actin may regulate the thermomorphogenesis of root through modulating intracellular auxin homeostasis.

In Arabidopsis, actin is divided into two subclasses: vegetative and reproductive subclasses [21]. Among the vegetative class actin, Actin 2 (*ACT2*) and Actin 8 (*ACT8*) belong to the subclass 1 with only one amino acid difference [25]. Although, *ACT2* mutant exhibits wavy root phenotype, lack of neither *ACT2* nor *ACT8* affect the primary root growth. For root hair elongation, they act redundantly [21]. On the other hand, *ACT7*, the lone member of subclass 2, regulates the primary root elongation. *act7-4* mutant exhibits dwarf phenotype at the seedling stage with twisted and short root [26]. *ACT7* is highly expressed in rapidly growing vegetative organs and is also required for seed germination [26,27]. Hence, we selected *ACT7* and *ACT8* actin isovariants with two unique characteristics from two different subclasses to check the role of actin isovariants in regulating root thermomorphogenesis under prolonged high temperature. Using a combinatorial approach of physiology and cell biology, here we demonstrate that subclass II actin isovariant *ACT7* plays a major role in regulating Arabidopsis roots thermoresponse through modulating PIN-mediated intracellular auxin homeostasis.

## 2. Results

### 2.1. Effect of Prolonged Moderate-High Temperature on Root Elongation 

Earlier, it was demonstrated that short treatment at moderate-high temperature (28–30 °C), stimulates root elongation and lateral root in Arabidopsis [16,17]. On the other hand, heat shock treatment (>40 °C) for short time period can lead to plant death [28]. However, effect of prolonged moderate-high temperature on root elongation remains obscure. 

To understand the effect of prolonged moderate-high temperature on root growth, we treated five days old seedlings of Columbia-0 (Col-0) at 29 °C for three days and compared the root elongation pattern against plants grown at 23 °C (Figure 1A,B). While at 23 °C, a steady increase in root growth was observed from day 1–3, at 29 °C on day 1, consistent with previous results, a stimulation in the root elongation was observed [16], but the root elongation became flat for subsequent days (Figure 1A,B). These results suggest that roots lose their normal elongation capability under prolonged moderate-high temperature treatment.

Primary root elongation relies on the coordinated function of cell elongation and meristematic cell division [23]. To provide a mechanistic explanation of high temperature induced alteration of root elongation, we performed the kinematic analysis by measuring root elongation rate and the length of newly mature cortical cells. The ratio of root elongation rate and cell length represents the output of the meristem (for that file of cortical cells), reflecting both the number of dividing cells and their rates of division [29]. Kinematic analysis revealed that prolonged high temperature results in reduced cell elongation and cell production in wild type (Table 1). These results suggest that unlike short-term moderate-high temperature treatment, long-term moderate-high temperature treatment inhibits progressive root growth, which is a combined effect of reduced cell elongation and cell production rate.

### 2.2. ACTIN7 Plays an Important Role in Modulating Primary Root Growth under Prolonged High Temperature

Actin cytoskeleton is a dynamic structure which forms filaments from globular monomer actin proteins that serves as a track for vesicle movement and regulates many developmental processes including root elongation [20,21,23]. Both the high and low temperatures affect protein trafficking in Arabidopsis root [16,29]. High temperature also modulates actin dynamicity in Arabidopsis root [30]. Since actin dynamicity can control protein trafficking and both the cellular processes are influenced by high temperature and affect root elongation in Arabidopsis, we investigated whether actin plays any role in regulating root elongation under prolonged high temperature. To test this, we used *act7-4* and *act8-2* mutants, representing each subclass of vegetative actin. Interestingly, we observed that *act7-4* root elongation is severely sensitive to prolonged moderate-high temperature treatment, while *act8-2* responded like wild-type. Initial exposure to high temperature resulted in a similar response in wild type and *act7-4*, and an increase in root elongation was observed on day 1 compared with the plants grown at 23 °C. (Figure 1A,C). In contrast to wild-type, where the root elongation became flat on day 2 and a slight reduction of root elongation was observed on day 3, *act7-4* root elongation was severely inhibited (Figure 1A,C). On the other hand, *act8-2* mutant showed similar root elongation response like wild-type under moderate-high temperature (Figure 1A,C). These results suggest that *ACT7* is required to combat the high temperature stress response in root. To further understand the consequence of loss of *ACT7* for root elongation under high temperature, we performed kinematic analysis. Consistent with their reduced root elongation, *act7-4* showed drastic reduction in cell production rate and cell elongation (Table 1). Collectively these results demonstrate that subclass II actin isovariant *ACT7* plays an important role in long-term moderate-high temperature response.

### 2.3. Long-Term Moderate-High Temperature Alters Intracellular Actin Organization in act7-4

Actin dynamicity is essential for normal plant growth and development. Previous reports suggest that high temperature alters the actin cytoskeleton structure and dynamicity depending on the intensity of the temperature and duration of the treatment. For instance, heat shock (45 °C for 45 min) resulted in strong actin filament depolymerization in Arabidopsis hypocotyl [31], heat stress (37 °C for 24 h) reorganized actin to transvers cables in hypocotyl, and heat stress at 35 °C for 6 h induced thick bundles of actin filaments in root epidermal cells of Arabidopsis seedling [30,31]. To find out the effect of prolonged high temperature stress on actin dynamicity, we performed live cell imaging using ABD2-GFP marker line where GFP is tagged with ACTIN BINDING DOMAIN 2 of fimbrin in wild type background and *act7-4* background using confocal microscope and quantified the actin assembly [32]. Typically, four different parameters are used for actin quantification, namely: (1) occupancy that represents filament density, (2) skewness that indicates filament bundling, (3) ∆θ that represents average angle against longitudinal axis, and (4) normAvgRad that represents parallelness of the filamentous actin [33,34]. Skewness and average angle of *act7-4* mutant is significantly different than wild-type at 23 °C, indicating that actin dynamicity is already altered in absence of *ACT7* in Arabidopsis root (Figure 2D,E).

Prolonged moderate-high temperature alters the filament parallelness and orientation in wild-type (Figure 2E,F). The effect of prolonged moderate-high temperature was found to be profound in *act7-4* mutant. More changes in filament parallelness and orientation were observed in *act7-4* compared to wild-type (Figure 2E,F). Prolonged moderate-high temperature treatment also induced more actin filament bundling in *act7-4* (Figure 2A,D). These data suggest that high temperature-mediated alteration of actin dynamicity is *ACT7* dependent and loss of *ACT7* makes the cellular actin more susceptible to high-temperature stress.

### 2.4. Prolonged Moderate-High Temperature Treatment Reduces the Abundance of Auxin Efflux Transporters PIN1 and PIN2

Directional flow of auxin facilitated by PINs and AUX/LAX transporters maintains root development [35,36,37]. Both *PIN1* and *PIN2* mediated auxin flow contributes in maintaining proper root length and meristem size [38]. Actin filament polymerization and depolymerization block *PIN1* trafficking in Arabidopsis root [22]. Latranculin B (LatB)-induced depolymerization of actin filaments produces intercellular agglomeration of *PIN2* [23]. Since prolonged moderate-high temperature affects the root elongation and also alters the cellular actin dynamicity, we therefore examined whether high temperature affects the localization or the expression of *PIN1* and *PIN2* in wild-type and *act7-4* mutant. Both the *PIN1* and *PIN2* abundance, but not the localization was affected by prolonged high temperature treatment (Figure 3 and Figure 4). Moderate-high temperature affects the abundance of both *PIN1* and *PIN2* in wild type and *act7-4*, albeit differentially. Quantitative data obtained from GFP fluorescence analysis show that the reduction of *PIN1* abundance at 29 °C is 46.5% and 33% in wild-type and *act7-4* background, respectively, compared to 23 °C (Figure 3). *PIN2* abundance was found to be more affected in the *act7-4* background. In wild type, the decrease in *PIN2* abundance was 35%, while in in *act7-4* it was 75% (Figure 4B). To confirm that the high temperature-induced alteration in PIN abundance is *ACT7* specific, we also checked *act8-2 PIN2*-GFP. As expected, no change was observed in *PIN2* abundance in *act8-2* mutant background comparing to wild-type under long-term moderate-high temperature treatment (Appendix A).

To understand whether the moderate-high temperature-induced reduction in PIN abundance is linked to the direct inhibition of gene expression, we compared the *PIN1* and *PIN2* expression under optimal and prolonged moderate-high temperatures. The expression analysis revealed that high temperature does not affect *PIN1* and *PIN2* at transcriptional level (Appendix A). Collectively, these results suggest that moderate-high temperature affects *PIN1* and *PIN2* at the translational level.

### 2.5. Moderate-High Temperature Treatment Does Not Affect the Expression of Auxin Influx Transporter AUX1

In contrast to PIN family proteins, which use a constitutive dynamic trafficking pathway [22,39], auxin influx carrier AUX1 shows a more stable plasma membrane localization [40]. It has also been shown that PIN and AUX1 proteins use distinct pathways for protein trafficking, while PINs use a BFA sensitive GNOM dependent trafficking pathway, membrane localized AUX1 trafficking is mediated by BFA resistant GNOM independent pathway [40]. However, like PINs, AUX1 intracellular trafficking was shown to be actin dependent. Using high concentration of actin depolymerizing drug LatB, it was shown that actin depolymerization leads intracellular agglomeration as well as reduced polar localization of AUX1 [40,41]. Since high temperature alters actin dynamicity, we checked whether long-term moderate-high temperature treatment alters the abundance or the trafficking of auxin influx transporter AUX1-YFP. Unlike PINs, AUX1 abundance was not altered in wild type or *act7-4* mutant under prolonged moderate-high temperature (Figure 5). These results suggest that prolonged moderate-high temperature specifically targets PIN proteins through actin isovariant *ACT7*.

### 2.6. Intracellular Auxin Distribution Is Altered by Long-Term Moderate-High Temperature Treatment

Intracellular auxin distribution in root largely relies on the polar transport of auxin mediated by PIN family proteins [42,43]. Since prolonged high temperature treatment alters the *PIN1* and *PIN2* abundance, it is expected that it will also affect the intracellular auxin distribution. To understand the end-result of long-term high temperature treatment on the auxin level in the root, we used auxin reporter line *AtIAA2::GUS*, where *IAA2* promoter is transcriptionally fused to *GUS* [44], in wild type and *act7-4* backgrounds. *AtIAA2::GUS* reporter line is strongly induced minutes after auxin exposure and a reliable reporter line to assess cellular auxin response [44]. Compared to 23 °C, *IAA2-GUS* expression is slightly increased in the vascular cylinder of the wild type root at 29 °C (Figure 6). At the same time, a depletion in auxin maximum was observed in root columella cells (Figure 6). These observations are consistent with the reduced *PIN1* and *PIN2* abundance under high-temperature treatment (Figure 3 and Figure 4). As observed earlier, loss of *ACT7* results in a dramatic reduction in auxin response [45]. Even at 23 °C, *act7-4* shows an extremely low level of GUS staining (Figure 6, [45]). Compared with 23 °C, prolonged high temperature treatment increased the auxin response both at columella cells and vascular tissues of *act7-4* (Figure 6), which is possibly due to the local accumulation of auxins resulted from the enhanced reduction of *PIN1* and *PIN2* abundance. To further confirm that the high-temperature-induced altered auxin response is specifically modulated by actin isovariant *ACT7*, we checked the auxin response in *act8-2* mutant using *act8-2 IAA2-GUS*. *act8-2* mutant showed wild-type like GUS staining pattern under moderate-high temperature treatment (Appendix A).

## 3. Discussion

Exposure of plants to moderate-high temperature results in various developmental changes that are largely regulated by plant hormone auxin [14,16]. Although a considerable amount of data is available for short-term effect of moderate-high temperature, the prolonged effect of moderate-high temperature on plant development remains obscure. In this work, we provided evidence that Arabidopsis root development under prolonged moderate-high temperature is modulated by subclass II vegetative actin isovariant *ACT7*. We also demonstrated that under prolonged high temperature, roots lose their normal elongation capability, which is linked to the intracellular auxin homeostasis regulated by vegetative class actin *ACT7*. Behavior of root elongation shows differential responses under moderate-high temperature for short and long exposures. In contrast to short exposure, where increase in the root elongation was observed, under the long exposure, root loses its normal elongation ability (Figure 1). This behavioral difference was found to be linked to the altered actin organization, auxin homeostasis and differential regulation of auxin efflux carriers *PIN1* and *PIN2*. Short exposure at high temperature redirects the auxin transporter *PIN2* to plasma membrane in SNX1 dependent manner, that increases the shootward auxin transport and consequently stimulates root elongation [16]. On the other hand, long exposure to moderate-high temperature inhibits the abundance of both *PIN1* and *PIN2* resulting in altered auxin distribution and slows down the root elongation process (Figure 3, Figure 4 and Figure 6).

We found that Actin isovariant *ACT7*, belonging to the subclass II vegetative class, plays an important role in modulating root thermal response. High temperature affects the actin organization drastically in the *act7-4* mutants (Figure 2). Compared with wild-type, three key actin parameters, filament bundling, filament parallelness, and filament orientation, were found to be significantly altered in *act7-4* mutant (Figure 2). Consistently, high temperature profoundly inhibited the cell length and cell division in *act7-4* (Table 1). This is consistent with previous findings as actin organization had been shown to be explicitly involved in regulating both cell elongation and cell division. For instance, in tobacco BY-2 cells, YFP-mTn-induced actin bundling inhibits cell division, and de-bundling of actin filaments by IAA or NAA treatment can reverse it [46]. In Arabidopsis root, 2,4-D and LatB-induced actin filament degradation impede both cell division and cell elongation but IAA and NAA-induced actin polymerization only restrains cell elongation [23]. 

We show that auxin efflux transporters *PIN1* and *PIN2* expression is reduced by prolonged moderate-high temperature. *PIN1* abundance was found to be more affected than *PIN2* in the wild-type background, while the inhibition was similar in *act7-4* background. These results raise the possibility that a high-temperature mediated effect on *PIN2* is more *ACT7* dependent. This idea is not inconsistent as *PIN1* and *PIN2* respond differentially towards various abiotic stresses. For instance, short-term high temperature exposure enhances the recycling of *PIN2* from vacuole to plasma membrane but did not affect *PIN1* [16], under low boron condition, accumulation of *PIN1* is reduced without affecting *PIN2* [47].

The reduced abundance of *PIN1* and *PIN2* under prolonged moderate-high temperature was also found to be linked with isovariant specific cellular actin organization. In agreement to our data, auxin inducive actin filament remodeling defective rice mutant *rmd-1* and *rmd-2* showed reduced plasma membrane localized *PIN2* with increased internalization [48]. Strigolactone analog GR24 increases the polar localization of *PIN2* in the membrane through reducing F actin bundling and enhancing actin dynamicity [49]. Under low phosphate condition, MAX2-dependent *PIN2* trafficking and polarization to the PM decreased due to the increase in actin bundling [50]. TIBA-induced polymerization of actin filaments also reduces plasma-membrane localization and enhanced lytic vacuolar accumulation of *PIN2* [51]. Rho GTPase ROP2 and effector protein RIC4 induces fine filamentous actin and increases polar localization by inhibiting *PIN1* endocytosis [52]. In contrast, it has been shown that actin filament stabilization through SPIKE1-ROP6-RIC1 inhibits *PIN2* internalization in response to auxin [53]. These results suggest that actin dynamicity governed by various regulators is influenced by environmental cues which can control *PIN1* and *PIN2* abundance.

Intriguingly, high temperature did not affect either the expression or the trafficking of AUX1. This is not inconsistent as it has been shown that PINs and AUX1 trafficking are regulated by two distinct regulatory pathways [40]. Internalization of AUX1 from plasma membrane has been shown to be actin dependent. A high concentration of actin depolymerizing drug LatB treatment resulted in intracellular agglomeration of AUX1 and also affected the polarity in protophloem cells [40,41]. In this study, we did not observe any change of AUX1 trafficking or localization in *act7-4* mutant, indicating that AUX1 trafficking is not *ACT7* dependent. This apparent discrepancy may be due to the fact that a high concentration of actin depolymerizing drug wipes out all the actin resulting in complete depletion of filamentous actin, while in *act7-4* mutant actin filaments from *ACT8* and *ACT2* isovariants still exist. This residual actin may be sufficient to stabilize AUX1 in the plasma membrane. Nevertheless, our results strongly suggest that AUX1 trafficking is *ACT7* independent.

No alteration of *PIN1* and *PIN2* expression under moderate-high temperature confirms that prolonged high temperature either inhibits PINs at translational level or boosts ubiquitinated degradation or both. In *act7-4* root, increased accumulation of internalized *PIN1* and *PIN2* was observed at 23 °C, which disappeared after long-term high temperature treatment. Reduced plasma membrane localization of *PIN1* and *PIN2* without internal accumulation supports the idea that the reduction of the PIN abundance is possibly due to inhibition of protein synthesis. Post translational regulation of proteins by various stresses is not uncommon. For instance, prolonged high temperature has been reported to promote the ubiquitinated degradation of brassinosteroid receptor BRI1 [54]. On the other hand, PIN-LIKES 6 is down regulated through post translational modification after long-term high-temperature treatment [55]. Cold stress down regulates SEC7 containing ARF-GEF GNOM at translational level in Arabidopsis [56]. Long-term (Cs^+^) metal stress reduces expression of cesium uptake transporters ABCG37 and ABCG33 without affecting their transcription [57]. E3 ubiquitin ligase IRT Degradation Factor 1 (IDF1) degrades ferrous Fe uptake transporter IRON-REGULATED TRANSPORTER1 (IRT1) depending on environmental conditions to maintain Fe homeostasis [58]. Although the detail mechanism is still unclear, these results suggest that translational regulation of proteins is an important regulatory mechanism for plant to combat various abiotic stresses.

Although *ACT7* and *ACT8* have high homology (7% difference at amino acid level), here we demonstrate that root thermomorphogenesis is linked to *ACT7* but not *ACT8*. This is possibly due to the subclass variance, and subtle changes in biochemical properties between *ACT7* and *ACT8* [59]. Arabidopsis vegetative actin subclass II is in a sister group of the subclass I clade in the phylogenetic tree. Subclass I differs by 7% amino acid from subclass II, and these classes of Arabidopsis actin are more closely related to the genes from other plant species than they are to each other. *ACT2/8* clade is closer to actin genes from maize and rice compared to *ACT7*, while *ACT7* is more similar to the genes from pea, carrot, potato, and pine than *ACT8*, suggesting that they originated from different lineages [59]. The differences between these two subclasses are also evident in their amino acid compositions. While *ACT2* and *ACT8* contain charged and polar residues at position Asp5 and Asn50, these residues are absent in *ACT7*. His41 and ser170 of subclass I are also replaced with Thr41 and Ala170 in subclass II. These subtle changes in amino acid compositions have the potential to alter actin–actin or actin–actin binding protein interactions [59]. Differential biochemical properties of *ACT2* and *ACT7* had already been reported in terms of polymerization and phosphate release rates. Actin binding proteins, profilin 1 and profilin 2 inhibit the polymerization of *ACT7* while the effect is minimal on *ACT2* [60]. In addition, the promoter region of these two subclasses also differ largely in their binding elements [27,61] (http://www.athamap.de/search_gene.php, accessed on 30 May 2021). These changes in amino acid residues, biochemical properties and promoter binding elements possibly contribute to the differential functional roles of subclass I and subclass II actin for plant development and high temperature response.

## 4. Materials and Methods

### 4.1. Plant Materials

All marker lines and mutant are in the Columbia background of *A. thaliana*. *PIN2*-GFP [62] was provided by Ben Scheres (University of Utrecht, Utrecht, the Netherlands), *PIN1*-GFP was obtained from ABRC. AUX1-YFP was a gift from Malcolm Bennett (University of Nottingham, Nottingham, UK) [63,64]. Auxin marker lines *IAA2-GUS* was described earlier [44]. For live cell imaging, the transgenic ABD2-GFP [32] was used. *act7-4* and *act8-2* were a gift from Rich Meagher (University of Georgia, Athens, GA, USA) and described earlier [21,65]. *PIN2*-GFP, ABD2-GFP, AUX1-YFP, and *IAA2-GUS* transgenic lines were generated by crossing, and F3 homozygous lines were used for microscopy observations. 

### 4.2. Growth Conditions

Surface-sterilized seeds were placed on modified Hoagland medium (Baskin and Wilson, 1997) containing 1% (*w*/*v*) sucrose and 1% (*w*/*v*) agar (Difco Bacto agar; BD Laboratories, Sparks, MD, USA) (http://www.bdbiosciences.com, accessed on 30 May 2021). Two days after stratification at 4 °C in the dark, plates were transferred to a growth chamber (NK System; LH-70CCFL-CT) at 23 °C under continuous white light at an irradiance of 80–90 µmol m^−2^ s^−1^. The seedlings were grown vertically for 5 days. For root growth assay, 5-day-old seedlings were transferred to new Hoagland plates and kept at 23 °C (NK System; LH-70CCFL-CT, Tokyo, Japan) and 29 °C growth chamber (NK System; LH-1-120.S, Tokyo, Japan) for 72-h treatment under continuous white light at an irradiance of 80–90 µmol m^−2^ s^−1^. Root growth elongation at different time point was carried out based on the procedure described earlier.

All phenotypic images were taken by Canon Power Shot A640 (Canon, Japan) without flash and using micro-focus function. Root elongation was measured using ImageJ (https://imagej.nih.gov/ij/, accessed on 30 May 2021) software.

### 4.3. Kinematic Analysis

The cell production assay was performed as described earlier [23]. Seedlings were grown vertically after stratification. Four-day-old seedlings were transferred to new medium. Cortical cell length was measured after 72 h. To ensure newly matured cells were scored, no cell was measured closer to the tip than the position where root hair length was roughly half maximal. The length of 10 mature cortical cells per root and eight roots used per treatment were measured. Cell production rate was calculated by taking the ratio of root elongation rate and average cell length for each genotype.

### 4.4. Chemicals

Cell clearing solution (Visikol) was purchased from Visikol Inc., Hampton, NJ, USA (https://visikol.com/, accessed on 30 May 2021). All other chemicals were purchased from Wako Pure Chemical Industries, Osaka, Japan.

### 4.5. GUS Staining

GUS staining was performed as described earlier [66]. In brief, 5-day-old seedlings were transferred to a new agar plate and grown vertically at 23 and 29 °C under continuous white light. After 72 h, *IAA2-GUS* and *act7-4/act8-2 IAA2-GUS* seedlings were transferred to GUS staining buffer (100 mM sodium phosphate, pH 7.0, 10 mM EDTA, 0.5 mM potassium ferricyanide, 0.5 mM potassium ferrocyanide, and 0.1% Triton X-100) containing 1 mM X-gluc and incubated at 37 °C in the dark for 1 h. For cell clearing, Visikol was used as per manufacturer’s instruction. The roots were imaged with a light microscope (Nikon Diaphot) equipped with a digital camera control unit [Digital Sight (DS-L2); Nikon, Tokyo, Japan (http://www.nikon.com/, accessed on 30 May 2021).

### 4.6. Quantification of GUS Staining

For quantification of GUS staining in root, the region of interest (ROI) was drawn and integrated intensity was measured for each seedling at HSB stack (saturated level) using the ImageJ software [67]. The intensity/area was measured for columella cells and vascular tissue in meristematic region which is in arbitrary unit.

### 4.7. Live-Cell Imaging

To image GFP, the transferred seedlings were incubated at 23 and 29 °C under continuous light for 72 h. After mounting on a large cover glass, the roots were imaged using a Nikon laser-scanning microscope (Eclipse Ti equipped with Nikon C2 Si laser-scanning unit) with a 20X/40X water objective. Fluorescence intensities were measured by drawing a ROI in the images (Approximately 250 µm from tip) obtained from live-cell imaging using Image J software 1.8.0.

### 4.8. Actin Quantification

Actin quantification was performed using ImageJ software 1.8.0 as described earlier [33]. Before measurements of actin parameters, the confocal images were skeletonized with using the ImageJ plug-in [34].

### 4.9. Gene Expression Analysis

RNA was extracted from the root tissue of 5-day-old vertically grown A. thaliana seedlings incubated at 23 and 29 °C for 72 h, using RNA Extraction Kit (QIAGEN, Hilden, Germany) with on-column DNA digestion to remove residual genomic DNA according to manufacturer’s protocol. Extracted RNA was tested for quality and quantity. Each RNA concentration was normalized with RNase-free water. Then, 1 µg of RNA was used to make 20 µL cDNA using qPCR RT kit, TOYOBO CO., LTD. Semi-quantitative RT–PCR was performed using 1 µL of cDNA. PCR conditions for *PIN1*: 95 °C—5 min, 95° C—30 s, 55 °C—30 s, 72 °C—30 s, 72 °C—10 min, 10 °C—hold, 30 cycles. PCR condition for *PIN2*: 95 °C—5 min, 95 °C—30 s, 59 °C—30 s, 72 °C—30 s, 72 °C—10 min, 10 °C—hold, 25 cycles. EF1α was used as control. Data were obtained from three biological replicates.

### 4.10. Statistical Analysis

Results are expressed as the means ± SE from the appropriate number of experiments as described in the figure legends. Two-tailed Students *t*-test or Tukey–Kramer multiple comparison tests were used to analyze statistical significance using R (https://www.r-project.org/, accessed on 30 May 2021).

## 5. Conclusions

In this study, we demonstrated that root thermomorphogeneis is regulated by subclass II specific actin isovariant *ACT7*, but not by the subclass I actin isovariant *ACT8*. Under moderate long temperature stress, *ACT7* plays a central role in maintaining optimal root elongation through regulating actin dynamicity and *PIN1* and *PIN2* abundance. The differential roles of *ACT7* and *ACT8* can be attributed to the changes in amino acid residues, biochemical properties and promoter binding elements. Further studies are required to clarify the mechanistic basis of how *ACT7* responds to high temperature.

## Figures and Tables

**Figure 1 ijms-22-07749-f001:**
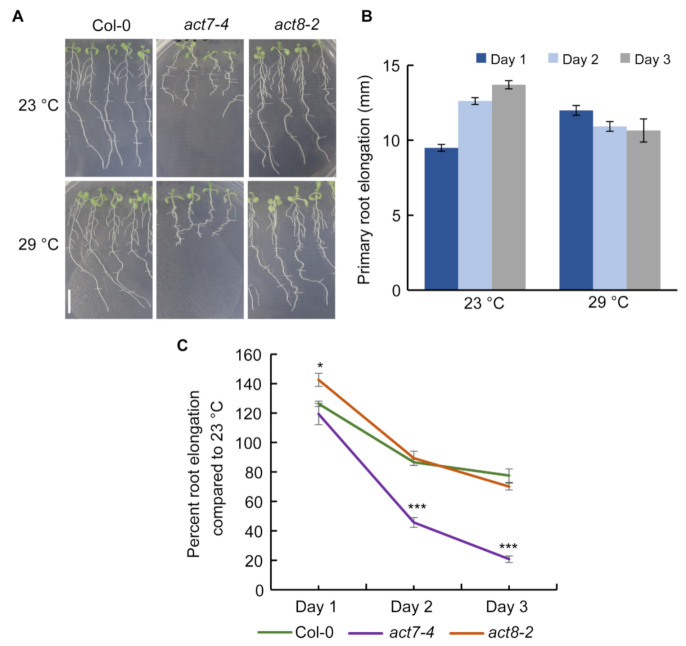
Effect of prolonged moderate-high temperature on root elongation of Col-0, *act7-4*, and *act8-2* mutants. (**A**) Phenotype of primary root elongation of Col-0, *act7-4*, and *act8-2* for 3 consecutive days at 23 °C and 29 °C. Root elongation was marked after every 24 h (Scale bar = 10 mm). (**B**) Primary root elongation of Col-0 for 3 consecutive days at 23 °C and 29 °C. (**C**) Primary root elongation of Col-0, *act7-4*, and *act8-2* in percentage for 3 consecutive days. Percentage root elongation at 29 °C is calculated against the root elongation at 23 °C for each genotype. Five-day-old seedlings were transferred to new agar plates and subjected to high-temperature treatment (29 °C) and control temperature (23 °C) for 3 days under light condition. Vertical bars represent mean ± SE of the experimental means from at least three independent experiments. Asterisks represent the statistical significance as judged by the Student’s *t*-test (* *p* < 0.05; *** *p* < 0.001).

**Figure 2 ijms-22-07749-f002:**
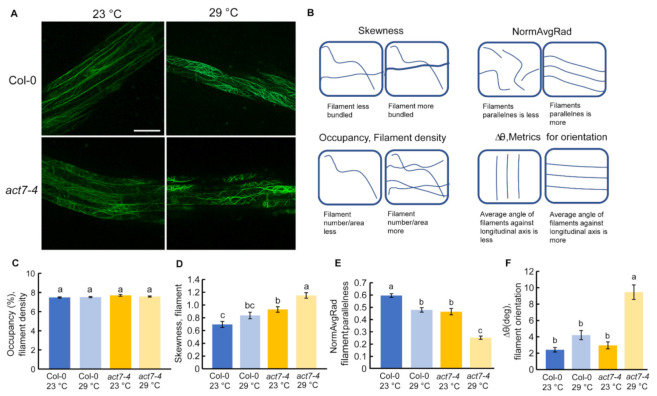
Effect of prolonged moderate-high temperature stress on cellular actin organization in Col-0 and *act7-4*. (**A**) Confocal images of ABD2-GFP and *act7-4* ABD2-GFP at 23 and 29 °C (Bars = 50 µm). (**B**) Schematic representation of quantified actin parameters. Quantification of actin filaments (**C**–**F**). (**C**) Percent occupancy, (**D**) Skewness, (**E**) NormAvgRad, and (**F**) ∆θ in degree. Five-day-old seedlings were transferred to new agar plates and subjected to high-temperature (29 °C) and control temperature (23 °C) treatment for 3 days under light condition. Vertical bars represent mean ± SE of the experimental means from at least three independent experiments (n = 3 or more), where experimental means were obtained from 30 to 50 cells. Comparisons between multiple groups were performed by analysis of variance (ANOVA) followed by the Tukey–Kramer test. The same letter indicates no significant differences (*p* < 0.05).

**Figure 3 ijms-22-07749-f003:**
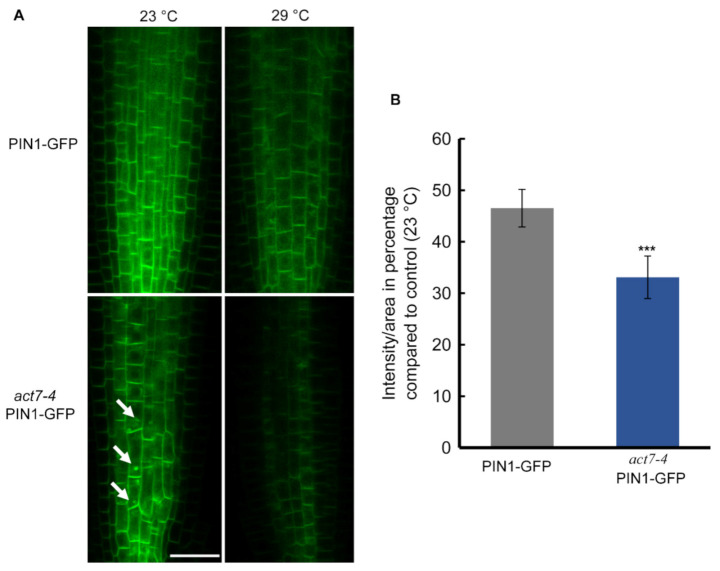
Prolonged moderate-high temperature affects *PIN1* abundance in roots of Col-0 and *act7-4* mutants. (**A**) *PIN1*-GFP and *act7-4 PIN1-GFP* at 23 and 29 °C. Bar = 25 µm. Arrowheads indicate the intracellular agglomeration of *PIN1* (**B**) Quantification of GFP fluorescence intensity from (**A**). The images were captured using the same confocal setting. Vertical bars represent mean ± SE of the 15–20 seedlings from at least three independent experiments. Asterisks represent the statistical significance between the means of the percentage Col-0 and mutant judged by the Student’s *t*-test (*** *p* < 0.001). Five-day-old light grown seedlings were transferred to new agar plates and subjected to high-temperature treatment (29 °C) for 72 h before imaged with confocal microscopy using the same confocal settings and are representative of 15–20 seedlings obtained from three independent experiments.

**Figure 4 ijms-22-07749-f004:**
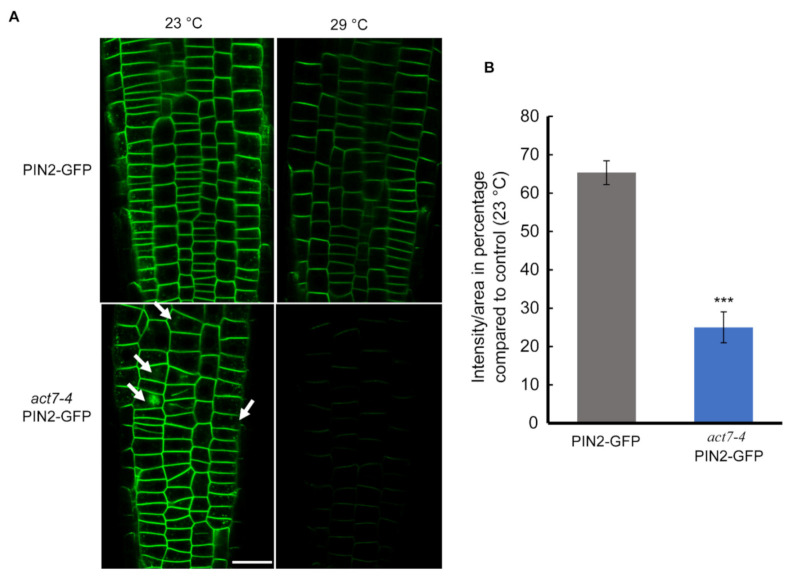
Prolonged moderate-high temperature diminishes *PIN2* abundance drastically in *act7-4* roots. (**A**) Cellular imaging of *PIN2*-GFP and *act7-4 PIN2-GFP* at 23 °C and 29 °C. Bar = 25 µm. Arrowheads indicate the intracellular agglomeration of *PIN2* (**B**) Quantification of GFP fluorescence intensity from (**A**). The images were captured using the same confocal setting. Asterisks represent the statistical significance between the means of the percentage Col-0 and mutant judged by the Student’s *t*-test (*** *p* < 0.001). Vertical bars in the graph represent mean ± SE of the 15–20 seedlings from at least three independent experiments. Five-day-old light grown seedlings were transferred to new agar plates and subjected to high-temperature treatment (29 °C) for 72 h before imaged with confocal microscopy using the same confocal settings. Images are representative of 15–20 seedlings obtained from three independent experiments.

**Figure 5 ijms-22-07749-f005:**
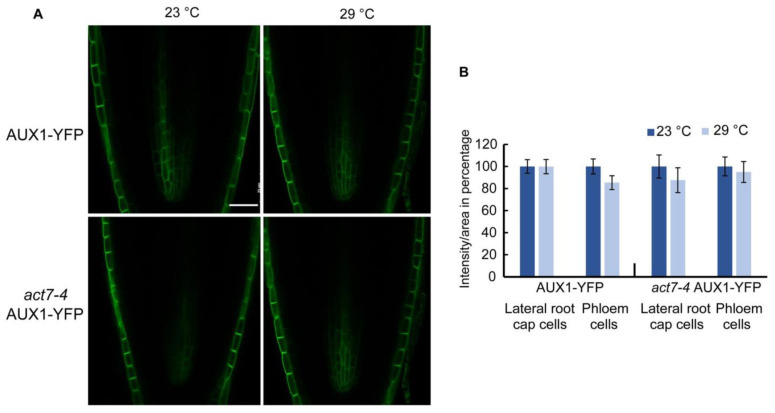
Prolonged high temperature does not alter AUX1 abundance in Col-0 and act7-4 mutants. (**A**) Cellular imaging of AUX1-YFP and *act7-4* AUX1-YFP at 23 and 29 °C. Bar = 25 µm. (**B**) Quantification of GFP fluorescence intensity from (**A**). Five-day-old light grown seedlings of AUX1-YFP and *act7-4* AUX1-YFP were transferred to new agar plates and subjected to high-temperature treatment (29 °C) for 72 h before imaged with confocal microscopy using the same confocal settings and are representative of 15–20 seedlings obtained from three independent experiments. Bar = 25 µm.

**Figure 6 ijms-22-07749-f006:**
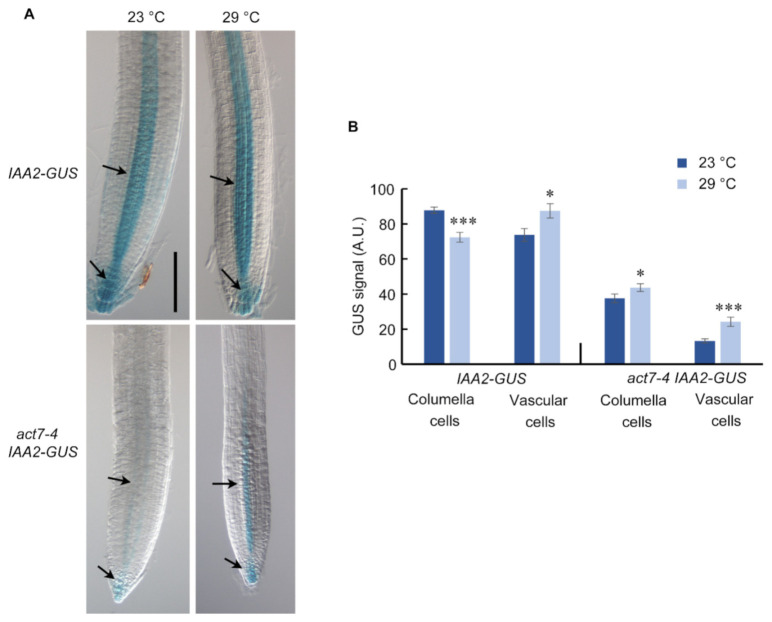
Prolonged high temperature alters the auxin response in Col-0 and *act7-4* roots. (**A**) *IAA2::GUS* and *act7-4XIAA2::GUS* at 23 and 29 °C. Five-day-old *IAA2::GUS* and *act7-4XIAA2::GUS* seedlings were transferred to 23 and 29 °C for 72 h. Seedlings were stained in a buffer containing 1 mM X-gluc for 1 h at 37 °C and cleared for photography. These are representative images stained in at least three separate experiments. Bar = 100 µm. (**B**) Quantification of GUS signal in the root from (**A**). Asterisks represent the statistical significance between the means for genotype specific treatment judged by the Student’s *t*-test (* *p* < 0.05 and *** *p* < 0.001). Vertical bars in the graph represent mean ± SE of the 15–20 seedlings from at least three independent experiments.

**Table 1 ijms-22-07749-t001:** Effect of moderate-high temperature stress on cell length and cell production rate in Col-0 and *act7-4* mutant.

Genotype	Condition	Elongation Rate (mm day^−1^)	Cell Length (µm)	Cell Production Rate (Cell Day^−1^)
Col-0	23 °C	10.17 ± 0.09	193.54 ± 1.72	52.59 ± 0.5
Col-0	29 °C	7.19 ± 0.33	181.22 ± 1.67 **	39.65 ± 1.48 **
*act7-4*	23 °C	6.03 ± 0.15	149.49 ± 0.24	40.34 ± 1.07
*act7-4*	29 °C	1.09 ± 0.063	82.82 ± 3.26 ***	13.23 ± 0.66 ***

Four-day-old seedlings were exposed to the control (23 °C) and high temperature (29 °C) for 3 days and the measurements reflect the behavior over the third day of treatment. Data are means ± SE of three replicate experiments. Asterisks represent the statistical significance between control and high temperature treatment in Col-0 and *act7-4* as judged by the Student’s *t*-test (** *p* < 0.01, *** *p* < 0.001).

## Data Availability

The data presented in this study are available upon request.

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
