# Peer review of "Actin Isovariant ACT7 Modulates Root Thermomorphogenesis by Altering Intracellular Auxin Homeostasis"

_ijms, 2021, doi:10.3390/ijms22147749_

Round 1

Reviewer 1 Report

Summary:

Auxin carrier proteins move along the actin cytoskeleton. High temperature is known to disrupt auxin carrier transport as well as actin dynamicity. Authors want to establish actin isovariants controlling the link between temperature, auxin trafficking, and development of the root. They therefore picked one isovariant from each subclass and tested role in thermomorphogenesis. ACT7 is shown to have a specific role in this process by regulating PIN1 and PIN2 accumulation and trafficking.

Comments:

  1. The rationale for why they looked specifically in the root is not laid out. Indeed, the big idea should be laid out better. For example, has effect of prolonged moderate vs short high temp not been looked at? Or has it never been looked at specifically in roots which is why they’re testing it? Or have the contributions of specific isovariants never been examined? The rationale for the specific experiments performed in this paper should be better laid out, in the Intro as well as beginning of results.
  2. The large paragraph of text in the Intro is difficult to read.
  3. PIN1 imaging needs to be redone. It is difficult to see the plasma membrane localization and impossible to see the polar localization. PIN2 imaging is better, however polar localization is also not evident. Higher magnification will resolve this.
  4. The PIN1 and PIN2 figures can be combined into one figure.
  5. It is not clear whether there are any PIN1/2 defects in the act7 mutant at 23C. This is because the data is presented as a ratio of 29 degrees to 23 degrees which obscures this information. Presenting the raw 23 degree data as well as the ratio would be helpful.
  6. PIN1/PIN2 turnover is speculated to occur either by translational inhibition or by ubiquitin-mediated turnover. This hypothesis is directly testable via MG132 protease inhibitors and should be done.
  7. Why have only PIN1 and PIN2 been examined? Are no other PINs potentially involved? Reporter lines exist for all PINs. Checking the effects of 23 degree vs 29 degree in these reporters can easily be done. No need to introgress act7 in my opinion. This will save the authors some time.
  8. One of the most intriguing findings of the paper is that AUX1 also affected by actin poly or depoly drugs, yet isn’t affected by heat or the act7 mutant. This is striking! This provides evidence of a specific relationship between ACT7, temperature, and PINs in altering auxin intracellular trafficking. This is actually quite puzzling…. So: does ABD2-GFP mark all actin or only actin containing ACT7? If it ABD2 does bind to all actin, then the actin imaging/measurements suggest the entire actin cytoskeleton has defects. How then do you get specific effects on PIN intracellular trafficking? Why isn’t AUX1 affected if it also utilizes the actin cytoskeleton for its trafficking? Need clarification and discussion on this.
  9. Analyses of cellular actin organization and comprehensive and done well.
  10. The IAA2 explanation and nomenclature is not clear. This is a reporter line (PIG4::GUS) in which the promoter of IAA2 is transcriptionally fused to GUS (Luschnig et al 1998). The authors need to explicitly state this and state that this reporter is strongly induced minutes after auxin exposure. “Auxin reporter line” is very vague given the number of auxin reporter lines and the range of auxin readouts they capture (e.g. DR5 vs R2D2 etc). For nomenclature, IAA2-GUS implies a translational fusion of the IAA2 protein to the GUS protein. This is confusing and should be clarified.
  11. Article is very well cited and sourced. This is a commendable and thorough review of the literature.

Author Response

Comments:

  1. The rationale for why they looked specifically in the root is not laid out. Indeed, the big idea should be laid out better. For example, has effect of prolonged moderate vs short high temp not been looked at? Or has it never been looked at specifically in roots which is why they’re testing it? Or have the contributions of specific isovariants never been examined? The rationale for the specific experiments performed in this paper should be better laid out, in the Intro as well as beginning of results.

Thanks for your comment. We have revised the introduction and added few sentences to strengthen the background of the study (line-65-67;83-84). We already mentioned about the studies where only short term high temperature treatment was used for root thermomorphogenesis studies and there is a gap in the literature about the long term treatment (line 68-84). We apologize that probably it was not clear enough for the reviewer and hence we changed the text according to the reviewer’s suggestion.

2. The large paragraph of text in the Intro is difficult to read.

We agree with the reviewer. In the revised version, we divided the large paragraph in multiple paragraphs.

3. PIN1 imaging needs to be redone. It is difficult to see the plasma membrane localization and impossible to see the polar localization. PIN2 imaging is better, however polar localization is also not evident. Higher magnification will resolve this.

We revised the figure according to reviewer’s suggestions and used zoomed pictures for better clarity. We also added arrows to mark the intracellular agglomeration of PIN proteins.

4. The PIN1 and PIN2 figures can be combined into one figure.

We prefer to keep them as separate figures.

5. It is not clear whether there are any PIN1/2 defects in the act7 mutant at 23C. This is because the data is presented as a ratio of 29 degrees to 23 degrees which obscures this information. Presenting the raw 23 degree data as well as the ratio would be helpful.

Yes, PIN1/2 trafficking is disturbed in act7 mutant background, but the expression is unaltered at 23C. Hopefully, the new zoom figures solve this issue. We also showed this data in a preprint (Numata et a., 2021 ref. 45) focusing on the role of actin in meristem development at optimal temperature.

6. PIN1/PIN2 turnover is speculated to occur either by translational inhibition or by ubiquitin-mediated turnover. This hypothesis is directly testable via MG132 protease inhibitors and should be done.

Thanks for the suggestion. We also thought about the experiment. However, this is not possible as we are looking at long term response. The initial response (first 24h) of the root to high temperature is completely different than the response at later stage. Initially, the root elongation is stimulated but after 24h the roots lose their ability to elongate. If we want to investigate ubiquitin mediated protein degradation, the plants need to be kept in MG132 for 3 days, which is literally impossible as long term MG132 treatment results in toxicity and inhibits the plant development. The MG132 approach works only for short term experiments. In fact, we already used this approach to show that cesium regulates the translation of ABCG33 and ABCG37 through regulating protein synthesis but not the ubiquitin dependent protein degradation (ref:57)

7. Why have only PIN1 and PIN2 been examined? Are no other PINs potentially involved? Reporter lines exist for all PINs. Checking the effects of 23 degree vs 29 degree in these reporters can easily be done. No need to introgress act7 in my opinion. This will save the authors some time.

Thanks for the suggestion. The reason we focused on PIN1 and PIN2 is because PIN1 and PIN2 are the major regulators of the auxin homeostasis linking primary root growth. PIN3, PIN4 or PIN7 does regulate root gravity response, lateral root development etc. but not the primary root elongation. pin3pin4pin7 triple mutants show wild-type like root elongation (performed the experiment in our lab, (Blilou et al., 2005 Supp. Figure 2) Hence, we do not think this experiment will provide us any new information.

8. One of the most intriguing findings of the paper is that AUX1 also affected by actin poly or depoly drugs, yet isn’t affected by heat or the act7 mutant. This is striking! This provides evidence of a specific relationship between ACT7, temperature, and PINs in altering auxin intracellular trafficking. This is actually quite puzzling…. So: does ABD2-GFP mark all actin or only actin containing ACT7? If it ABD2 does bind to all actin, then the actin imaging/measurements suggest the entire actin cytoskeleton has defects. How then do you get specific effects on PIN intracellular trafficking? Why isn’t AUX1 affected if it also utilizes the actin cytoskeleton for its trafficking? Need clarification and discussion on this.

This is an interesting comment. AUX1 is a much more stable PM protein compared with PIN1 and PIN2. We also mentioned clearly that these two groups of proteins use distinct pathways for protein trafficking. Hence, it is not inconsistent that PINs and AUX1 respond differentially to high temperature. Internalization of AUX1 from plasma membrane has been shown to be actin dependent. High concentration of actin depolymerizing drug LatB treatment resulted in intracellular agglomeration of AUX1 and also affected the polarity in protophloem cells. In this study, we did not observe any change of AUX1 trafficking or localization in act7-4 mutant, indicating that AUX1 trafficking is not ACT7 dependent. This apparent discrepancy may be due to the fact that high concentration actin depolymerizing drug wipes out all the actin resulting in complete depletion of filamentous actin, while in act7-4 mutant actin filaments from ACT8 and ACT2 isovariants still exist. This residual actin may be sufficient to stabilize AUX1 in the plasma membrane. Nevertheless, our results strongly suggest that AUX1 trafficking is ACT7 independent. ABD2-GFP mark all actin.

To make this part more clear, we revised the result section (line 270-275) and also added a paragraph in the discussion (Line 379-391). Hope this will satisfy the reviewer.

9. Analyses of cellular actin organization and comprehensive and done well.

We appreciate reviewer’s comment.

10. The IAA2 explanation and nomenclature is not clear. This is a reporter line (PIG4::GUS) in which the promoter of IAA2 is transcriptionally fused to GUS (Luschnig et al 1998). The authors need to explicitly state this and state that this reporter is strongly induced minutes after auxin exposure. “Auxin reporter line” is very vague given the number of auxin reporter lines and the range of auxin readouts they capture (e.g. DR5 vs R2D2 etc). For nomenclature, IAA2-GUS implies a translational fusion of the IAA2 protein to the GUS protein. This is confusing and should be clarified.

We apologize for raising the confusion and completely agree with the reviewer. We have changed the text (line 295-298) and figure legend accordingly. Thank you for pointing out this.

11. Article is very well cited and sourced. This is a commendable and thorough review of the literature.

Thanks for the appreciation.

Reviewer 2 Report

The manuscript by Sumaya Parveen and Abidur Rahman characterize the role of Actin 7 during root response to high temperatures. Authors observed root growth, actin localization, abundance of auxin transporters (PINs and AUX1) and auxin reporter IAA2 during normal and elevated temperatures, thus showing the involvement of Actin 7 in the regulation of thermomorphogenesis. Overall, the manuscript is well written.

Here are some comments for the authors:

Introduction - authors should make sure that all the gene and protein names are fully introduced properly, first time they are mentioned in the text

Line 74 - authors used an example of protein misfolding during high temperature treatment using a reference from animal field. Is there no such observation coming from plant field?

Line 114 - the statement " and slight inhibition in the growth was observed on day 3". There is no inhibition, the graph in Fig.1B shows root growth at constant scale at 29 °C compared to the increase of root growth at room temperature. Inhibition is rather strong word in this case, I would suggest to the authors to rephrase their statements. Is there a significant difference between in root growth at the Day 1, 2 and 3 at 29 °C?
In the graph of Fig.1B, what is the star above 3 day at 29 °C referring to? It is not clearly stated in the figure legend what was sort of comparison have been done here.

Line 181 - Would it be possible for authors to add a schematic of cell with actin filaments and visualize the quantified parameters (occupancy, skewness, and others) for better understanding of the process. There is still some space in Fig. 2, where it could be placed.

Figure 2 - It is a bit hard to see the differences in the actin filaments on the images. Would it be possible to provide higher magnification images?

Line 196 - Filament orientation is not significantly altered in WT at 29 °C.

Line 201 - The introduction of Fig.2A in the text should be done earlier in the paragraph, not just at the end after introducing all the other panels of the figure.

Line 231 - Authors speculate that high temperatures are affecting PIN1 and PIN2 translation level. What about protein degradation?
If actin filaments are altered during high temperatures, is it possible that PINs are just being accumulated somewhere along the secretion way? In the ER, GA, TGN? Or are they just being degraded faster and they end up in vacuole?

Figures 3 and 4 - It would be better if authors provide higher magnification images as it is difficult to observe the subcellular localization of the PIN proteins. Moreover, in line 360 in the discussion, authors are stating "a huge accumulation of internalized PIN1 and PIN2 at 23 °C". This is impossible to observe and evaluate on the images provided.

Line 238-241 and 248-251 - What are these parts of text referring to? Is it just some sort of leftover text?

Line 276 - Why did authors decided to use IAA2 as an auxin reporter? Wouldn't it be more convenient to just use a standard DR5 or DII reporter?
Moreover, using a fluorescent auxin reporter would be easier for intensity quantification.

Line 278 - Authors stated that "a depletion in auxin maximum was observed in collumela cells". This was never shown, only indirectly by IAA2-GUS

Line 285 - Can the local accumulation of auxin results from local auxin biosynthesis in the root cells as published previously (DOI:https://doi.org/10.1016/j.devcel.2018.09.022)?

Figure 5 - The quantification of signal intensity is missing.

Figure 6 - Authors are not showing auxin distribution. It is just IAA2 expression pattern visualized by GUS, not auxin distribution (for that you would need to use DR5, DII or R2D2 markers). Please correct the wrong statements.
Further, I cannot locate the scale bar on the images in Fig. 6A.

Line 337/338 - I do not understand that sentence.

Line 359 - Authors stated that "prolonged high temperature either inhibits PINs at translational level or boost ubiquitinated degradation". Would it be possible for authors to dissect this question? To check the amount of protein degradation after temperature treatment or to use published non-ubiquitinated PIN2 protein?
Overall this part of the discussion is a bit confusing, authors are somehow more focusing on translation repression of PINs, while there are already quite some publications depicting the enhanced ubiquitin-mediated degradation after various treatments. I would suggest performing some additional experiment to make it clearer which one of these two options would it be.

Author Response

Here are some comments for the authors:

Introduction - authors should make sure that all the gene and protein names are fully introduced properly, first time they are mentioned in the text

Thanks for pointing it out. We revised the text accordingly.

Line 74 - authors used an example of protein misfolding during high temperature treatment using a reference from animal field. Is there no such observation coming from plant field?

Unfortunately, we could not find any such example from plant field. In fact, very few works have been done on this topic.

Line 114 - the statement " and slight inhibition in the growth was observed on day 3". There is no inhibition, the graph in Fig.1B shows root growth at constant scale at 29 °C compared to the increase of root growth at room temperature. Inhibition is rather strong word in this case, I would suggest to the authors to rephrase their statements. Is there a significant difference between in root growth at the Day 1, 2 and 3 at 29 °C?
In the graph of Fig.1B, what is the star above 3 day at 29 °C referring to? It is not clearly stated in the figure legend what was sort of comparison have been done here.

We apologize for the mistake. We agree with the reviewer comment. In the revised version, we corrected it (line 121, Figure 1).

Line 181 - Would it be possible for authors to add a schematic of cell with actin filaments and visualize the quantified parameters (occupancy, skewness, and others) for better understanding of the process. There is still some space in Fig. 2, where it could be placed.

Thanks for the suggestion. We added it as per reviewer’s suggestion (Figure 2B).

Figure 2 - It is a bit hard to see the differences in the actin filaments on the images. Would it be possible to provide higher magnification images?

This issue is linked with the reduced image quality in the word format. We are providing high resolution image which should solve the issue.

Line 196 - Filament orientation is not significantly altered in WT at 29 °C.

We apologize for the mistake. Corrected it in the revised version.

Line 201 - The introduction of Fig.2A in the text should be done earlier in the paragraph, not just at the end after introducing all the other panels of the figure.

We are not sure what the reviewer wanted to mean. Could not follow the comment.

Line 231 - Authors speculate that high temperatures are affecting PIN1 and PIN2 translation level. What about protein degradation?
If actin filaments are altered during high temperatures, is it possible that PINs are just being accumulated somewhere along the secretion way? In the ER, GA, TGN? Or are they just being degraded faster and they end up in vacuole?

Thanks for the comment. There are several possibilities. Because of long term incubation, it is difficult to conduct experiment with chemical inhibitors. However, the typical intracellular accumulation of PINS in act7-4 mutant at 23C disappeared at 29C. If the reduction is through the protein degradation, one would expect more accumulation of proteins in the vacuole. However, we did not observe any PIN accumulation in the vacuole.

Figures 3 and 4 - It would be better if authors provide higher magnification images as it is difficult to observe the subcellular localization of the PIN proteins. Moreover, in line 360 in the discussion, authors are stating "a huge accumulation of internalized PIN1 and PIN2 at 23 °C". This is impossible to observe and evaluate on the images provided.

We completely agree with the reviewer and changed the figures accordingly. We also used arrows to highlight the changes (New Figure 3, Figure 4).

Line 238-241 and 248-251 - What are these parts of text referring to? Is it just some sort of leftover text?

This is a part of the figure legend. It looks odd because of the formatting. Corrected it.

Line 276 - Why did authors decided to use IAA2 as an auxin reporter? Wouldn't it be more convenient to just use a standard DR5 or DII reporter?
Moreover, using a fluorescent auxin reporter would be easier for intensity quantification.

Reviewer 1 also raised the same question and we explained the reason for using IAA2-GUS in the revised text (line 295-298). IAA2-GUS is one of the most sensitive auxin response markers, although it is under used. It is far better than DR5.

Line 278 - Authors stated that "a depletion in auxin maximum was observed in collumela cells". This was never shown, only indirectly by IAA2-GUS

That is correct. The depletion of auxin maximum was inferred from the IAA2-GUS data and we believe that this is a standard procedure.

Line 285 - Can the local accumulation of auxin results from local auxin biosynthesis in the root cells as published previously (DOI:https://doi.org/10.1016/j.devcel.2018.09.022)?

There is a possibility but for this work, we believe the disruption of transport results in local accumulation of auxin.

Figure 5 - The quantification of signal intensity is missing.

We have added the quantification in the revised version (new Figure 5).

Figure 6 - Authors are not showing auxin distribution. It is just IAA2 expression pattern visualized by GUS, not auxin distribution (for that you would need to use DR5, DII or R2D2 markers). Please correct the wrong statements.
Further, I cannot locate the scale bar on the images in Fig. 6A.

We partially agree with the reviewer on this point. DR5 is a synthetic marker, how it is more reliable than IAA2 which is a native marker? DII and IAA2 are similar type of markers. However, we changed the distribution to response as this term is more appropriate than auxin distribution.

Line 337/338 - I do not understand that sentence.

Sorry for the confusion. We revised the sentence and corrected it.

Line 359 - Authors stated that "prolonged high temperature either inhibits PINs at translational level or boost ubiquitinated degradation". Would it be possible for authors to dissect this question? To check the amount of protein degradation after temperature treatment or to use published non-ubiquitinated PIN2 protein?
Overall this part of the discussion is a bit confusing, authors are somehow more focusing on translation repression of PINs, while there are already quite some publications depicting the enhanced ubiquitin-mediated degradation after various treatments. I would suggest performing some additional experiment to make it clearer which one of these two options would it be.

Using non-ubiquinated PIN2 is a great idea. However, this is out of scope of this paper as it will take long time to do the experiments. We also clearly mentioned that future work is necessary to disentangle this issue. This phenomenon is interesting and needs further work, which will be another whole new story.

We also thought about MG132 experiment. However, this is not possible as we are looking at long term response. The initial response (first 24h) of the root to high temperature is completely different than the response at later stage. Initially, the root elongation is stimulated but after 24h the roots lose their ability to elongate. If we want to investigate ubiquitin mediated protein degradation, the plants need to be kept in MG132 for 3 days, which is literally impossible as long term MG132 treatment results in toxicity and inhibits the plant development. The MG132 approach works only for short term experiments. In fact, we already used this approach to show that cesium regulates the translation of ABCG33 and ABCG37 through regulating protein synthesis but not the ubiquitin dependent protein degradation. This article was recently published in Molecular Plant (ref:57).

We would like to focus on this issue in future work.

Round 2

Reviewer 1 Report

The authors have addressed my concerns and the figures are noticeably improved.

Reviewer 2 Report

I would like to thank authors for considering my comments. Nevertheless, I would still encouraged them to use the term auxin concentration with caution, as they are using indirect tools visualizing the auxin transcriptional response.